# Effect of Upregulation of Transcription Factor TFDP1 Binding Promoter Activity Due to *RBP4* g.36491960G>C Mutation on the Proliferation of Goat Granulosa Cells

**DOI:** 10.3390/cells11142148

**Published:** 2022-07-08

**Authors:** Yufang Liu, Siwu Guo, Xiaoyun He, Yanting Jiang, Qionghua Hong, Rong Lan, Mingxing Chu

**Affiliations:** 1Key Laboratory of Animal Genetics, Breeding and Reproduction of Ministry of Agriculture and Rural Affairs, Institute of Animal Science, Chinese Academy of Agricultural Sciences, Beijing 100193, China; aigaiy@126.com (Y.L.); jibizhan19700814@163.com (S.G.); hexiaoyun@caas.cn (X.H.); 2Yunnan Animal Science and Veterinary Institute, Kunming 650224, China; jiangyanting-2007@163.com (Y.J.); yxh7168@126.com (Q.H.); rtlankitty@163.com (R.L.)

**Keywords:** goat, fertility, *RBP4*, *TFDP1*, granulosa cell proliferation

## Abstract

Retinol-binding protein 4 (RBP4), a member of the lipocalin family, is a specific carrier of retinol (vitamin A) in the blood. Numerous studies have shown that *RBP4* plays an important role in mammalian embryonic development and that mutations in *RBP4* can be used for the marker-assisted selection of animal reproductive traits. However, there are few studies on the regulation of reproduction and high-prolificacy traits by *RBP4* in goats. In this study, the 5′ flanking sequence of *RBP4* was amplified, and a G>C polymorphism in the promoter region -211 bp (g.36491960) was detected. An association analysis revealed that the respective first, second and third kidding number and mean kidding number of nanny goats with CC and GC genotypes (2.167 ± 0.085, 2.341 ± 0.104, 2.529 ± 0.107 and 2.189 ± 0.070 for CC and 2.052 ± 0.047, 2.206 ± 0.057, 2.341 ± 0.056 and 2.160 ± 0.039 for GC) were significantly higher (*p* < 0.05) than those with the GG genotype (1.893 ± 0.051, 2.027 ± 0.064, 2.107 ± 0.061 and 1.74 ± 0.05). The luciferase assay showed that luciferase activity was increased in C allele individuals compared with that in G allele individuals. A competitive electrophoretic mobility shift assay (EMSA) showed that individuals with the CC genotype had a stronger promoter region binding capacity than those with the GG genotype. In addition, transcription factor prediction software showed that the *RBP4* g.36491960G>C mutation added a novel binding site for transcription factor DP-1 (*TFDP1*). RT–qPCR results showed that the expression of *TFDP1* was significantly higher in the high-prolificacy group than in the low-prolificacy group, and the expression of *RBP4* was higher in both the CC and GC genotypes than that in the GG genotype. *TFDP1* overexpression significantly increased the expression of *RBP4* mRNA (*p* < 0.05) and the expression of the cell proliferation factors cyclin-D1, cyclin-D2 and *CDK4* (*p* < 0.05). The opposite trend was observed after interference with *TFDP1*. Both the EdU and CCK-8 results showed that *TFDP1* expression could regulate the proliferation of goat ovarian granulosa cells. In summary, our results showed that *RBP4* g.36491960G>C was significantly associated with fecundity traits in goats. The g.36491960G>C mutation enhanced the transcriptional activity of *RBP4* and increased the expression of *RBP4*, thus improving the fertility of Yunshang black goats.

## 1. Introduction

Goats are an economically important animal for daily human consumption, providing meat, milk, skins, and wool [1]. Yunshang black goats, with the advantages of prolificacy and fast growth, are an excellent local breed in the Yunnan Province of China, and the number of kids ranges from one to four per litter [2]. This high-fertility goat breed is of great value for both research and production. The kidding number is the most important factor in measuring the reproductive performance of goats [3]. Currently, great progress has been made in the study of reproductive traits in goats, and *PDGFRB, MARCH1, KDM6A, CSN11S, SIRT3, KITLG, GHR, ATBF1, INHA, GNRH1,* and *GDF9* have been identified as potential candidate genes for reproductive traits in goats [4,5,6,7,8,9,10,11,12,13,14]. *FER1L4* and *SRD5A2* may be associated with high fecundity in goats [15]. The kidding number is a complex quantitative trait regulated by the interaction of multiple genes. However, the main effector genes regulating high fertility in goats are still largely unknown. To better understand the genetic mechanisms underlying fertility in goats, it is essential to study the combined effects of multiple genes or loci on fertility. Therefore, there is a need to identify more relevant candidate genes, elucidate their molecular mechanisms of action, and refine the regulatory networks associated with fertility in goats [16].

Retinol, also known as vitamin A, functions through the binding and transport of its carrier proteins, the retinol binding proteins (RBPs). Retinol is involved in various reproductive processes in living organisms, including steroidogenesis, follicular development, oocyte maturation and early embryonic development, and is a key nutrient required for the normal development of many tissues [17,18,19,20]. Retinol-binding protein 4 (*RBP4*) belongs to the family of lipid transport proteins whose main function is to transfer retinol from the liver and adipose tissue to the blood and other tissues [21,22]. It functions by binding to retinol in the blood to form the transient RBP4-ROH complex, which is an important player that assists retinol in performing its physiological functions [23]. RBP4, which is involved in retinol metabolism, is also associated with the risk of many metabolic diseases such as insulin resistance (IR), type 2 diabetes, obesity, and cardiovascular risk diseases [24,25]. The expression of *RBP4* in adipose tissue is regulated by 17-β-oestradiol [26]. RBP4 can drive ovarian cancer cell migration and proliferation through RhoA/Rock1 and extracellular signal-regulated kinase pathways, which are involved in the expression of matrix metalloproteinase (MMP) 2 and MMP9 [27]. In goats, RBP4 is located on chromosome 26, and consists of six exons and encodes a protein containing 203 amino acids. Several studies have now shown that *RBP4* is expressed during the critical period of porcine gestation, plays an important role in embryonic development, and can significantly increase litter size in commercial pigs [28,29,30,31].

Moreover, the epigenome, comprising different mechanisms such as DNA methylation, remodeling, histone tail modifications, chromatin microRNAs and long noncoding RNAs, interacts with environmental factors such as nutrition, pathogens, and climate to influence the expression profile of genes and the emergence of specific phenotypes [32,33]. Multilevel interactions between the genome, epigenome and environmental factors may occur [34]. Furthermore, numerous lines of evidence suggest the influence of epigenomic variation on health and production [32,33,35]. The expression of eukaryotic genes is temporarily and multidimensionally controlled [36]. Only a relatively small portion of the entire genome is expressed in each tissue type, and the expression of genes depends on the stage of development [37]. Therefore, gene expression in eukaryotes is specific to each tissue [38]. Additionally, the number of gene products that are made in each tissue regulates the expression of that gene [39]. One of the basic activities in domestic animals is the study of genes and proteins related to economic traits at the cellular or chromosomal level [40]. In this study, we investigated the SNPs in the 5′ regulatory region of *RBP4*, determined the relationship between the gene and kidding number in Yunshang black goats by association analysis, and further explored the potential molecular mechanism of *RBP4*. All these data are important for elucidating the molecular characteristics of RBP4 and understanding the molecular mechanisms of how RBP4 regulates high fecundity in Yunshang black goats.

## 2. Materials and Methods

### 2.1. Ethics Statement

All experimental procedures involved in this study were approved by the Animal Welfare Division of the Institute of Animal Science, Chinese Academy of Agricultural Sciences (IAS-CAAS) (Beijing, China). Ethical approval was provided by the animal ethics committee of IAS-CAAS (No. IAS2021-25).

### 2.2. Animal Sample Collection

Blood samples were obtained from 400 female Yunshang black goats (the experimental animals were all unrelated individuals) aged two to five years old in Honghe Hani and Yi Autonomous Prefecture in the Yunnan Province of China; all experimental subjects were kept under the same conditions. Blood was collected from the jugular vein (10 mL/sample) using EDTA-K2 anticoagulation tubes and then stored at −20 °C. Ovarian tissues were obtained from slaughtered female goats of three different RBP4 genotypes (3 individuals per group) based on the results of genotype sequencing. All experimental goats used for molecular experiments were euthanized. Female goats were divided into two groups based on phenotypic data: low-prolificacy (an average kidding number of <2) and high-prolificacy groups (an average kidding number of ≥2); three healthy Yunshang black goats were randomly selected from each group for ovarian tissue collection. Immediately after collection, ovarian tissues were placed in liquid nitrogen and transferred to the laboratory, where they were stored at −80 °C.

### 2.3. DNA Extraction and Sequencing

DNA was extracted from 400 blood samples using a DNA extraction kit (Tiangen Biotechnology Co., Ltd., Beijing, China). The steps are described in the instructions of the Tiangen ‘Blood/Cell/Tissue Genomic DNA Extraction Kit’ (catalogue number: DP304). To prevent degradation, agarose gel electrophoresis and UV spectrophotometry were used to assess the quality and concentration of DNA after storage at −20 °C. PCR amplification of the upstream promoter region of RBP4 was performed to screen for SNPs that may affect the binding site of the RBP4 transcription factor. Primers were designed based on GenBank number NC_030833.1 from the National Center for Biotechnology Information (NCBI) database. A PCR was performed using a 20 μL mix containing 1 μL DNA sample (diluted to 5–50 ng/μL), 10 μL 2× Taq Master Mix (Novozymes Biotechnology Co., Ltd., Nanjing, China), 8 µL ddH_2_O and 0.5 μL 10 nmol/L each of upstream and downstream primers (Table 1). The amplification conditions were as follows: 95 °C for 4 min; 35 cycles of 95 °C for 30 s, annealing temperature for 60 s, and 72 °C for 30 s; and 72 °C for 5 min. The PCR products were sequenced by Sanger sequencing using the same primers as the amplification primers at Beijing Tianyi Huiyuan Biotechnology Co., Ltd. (Beijing, China).

### 2.4. Genotype Sequencing

*RBP4* g.36491960G>C was genotyped in 400 female goats using the SNP-KASP genotyping technique. The primers and probes used are shown in Table 2, and the reaction system and PCR procedure are shown in Appendix A Appendix A, respectively. Finally, the genotyping results were obtained by fluorescence quantitative PCR (ABI, 7900) using the experimental procedures of the ABI 7900 HT Fast Real Time PCR system.

### 2.5. Statistical and Bioinformatics Analysis

The allele frequency, heterozygosity and polymorphism information content were calculated using Excel. The association analyses between different genotypes and kidding numbers of Yunshang black goats were performed according to Li et al. [41]. The statistical analysis was performed using SPSS 19.0 statistical software (SPSS Inc., Armonk, NY, USA) for independent samples *t*-tests. All results are reported as the mean ± SEM.

The SNP sequence of *RBP4* was analysed using DNAstar software (DNAstar Inc., Madison, WI, USA). The core promoter of *RBP4* was predicted by Promoter 2.0 (https://services.healthtech.dtu.dk/service.php?Promoter-2.0, accessed on 20 December 2021), and the transcriptional binding site of the promoter was predicted by JASPAR (https://jaspar.genereg.net/, accessed on 5 January 2022).

### 2.6. Total RNA Extraction, cDNA Reverse Transcription and RT–qPCR

Total RNA was extracted from samples (tissue or cell) using TRIzol reagent according to the manufacturer’s instructions (TaKaRa Biotechnology Dalian Co., Ltd., Dalian, China). The quality and concentration of RNA were assessed by agarose gel electrophoresis and UV spectrophotometry, respectively. A total of 1 μg of RNA was used for reverse transcription (TaKaRa, Dalian, China). The synthesized cDNA was stored at −20 °C.

The goat *RBP4* sequence obtained from the NCBI database (GenBank NC_030833.1) was used to design primers for RT–qPCR (Table 3), and the goat ribosomal protein L19 (*RPL19*) (GenBank NC_030826.1) was used as a reference gene. According to the genotyping results, three genotypes existed for *RBP4* g.36491960G>C: wild-type pure homozygote GG, mutant heterozygote GC and mutant pure homozygote CC. The difference in *RBP4* expression between different genotypes of *RBP4* promoter region SNPs was verified for the three groups with three biological replicates each. RT–qPCR was performed using a TaKaRa SYBR premixed Ex Taq II kit (TaKaRa Bio). The 20 µL reaction system included 2.0 µL cDNA template, 10 µL SYBR PremixExTaq, 0.8 µL each forward and reverse primers, and 6.4 µL ddH_2_O. Three replicates of each sample were performed. The RT–qPCR protocol was as follows: 40 cycles of 95 °C for 30 s, 95 °C for 5 s, and 60 °C for 30 s.

The relative expression of genes was calculated using the 2*^−^*^ΔΔ*Ct*^ method [42]. First, the Ct values of the target genes were normalized to the Ct values of the internal reference genes for all test samples and calibration samples. The calibration sample was a mixture of all individual sample cDNAs. Second, the ΔCt values of the calibration samples were used to normalize the ΔCt values of the test samples. Finally, the expression level ratio was calculated. The calibration sample was a mixture of all individual cDNAs.

### 2.7. Electrophoretic Mobility Shift Assay (EMSA)

EMSA was performed according to the instructions in the competitive EMSA kit (Thermo Fisher, 20148, USA). Briefly, 20 µL of the reaction system was mixed with 2 µg of nuclear extract in the presence of 2 µL binding buffer (10×), 1 µL poly (dI. dC), 1 µL 50% glycerol, 1 µL 1% NP-40, 1 µL 100 mM MgCl_2_, 1 µL 200 mM EDTA and 2 µL labelled probe (except the negative control) and then supplemented with DEPC water to make 20 µL. The mixture was allowed to incubate at 20 °C for 25–30 min. The samples were mixed with 4 μL of 6× loading buffer on a 5.5% nondenaturing polyacrylamide gel for electrophoresis (1 h pre-electrophoresis had been performed prior to electrophoresis); the electrophoresis conditions were 150 V for 60 min. After completion, the electrophoresis was then rotated at 300 mA for 30 min. Crosslinking was performed by exposure at 20 cm under a UV lamp for 20 min. At the end of the process, the antibody was placed in a closure solution for 20 min and reacted with a 300-fold dilution for 30 min. Finally, the elution was coupled with a chemiluminescent solution, and imaging showed the binding bands of DNA to transcription factors.

### 2.8. Vector Construction

To determine the effect of the *RBP4* g.36491960G>C mutation site on its transcriptional efficiency, the promoter region of the GG genotype containing the *TFDP1* binding site +/− 100 bp was amplified and cloned into pGL3-basic to obtain pGL3-G. The promoter region containing the CC genotype was amplified +/− 100 bp and cloned into pGL3-basic to obtain pGL3-C.

For subsequent validation experiments of *RBP4* versus proliferation factors in the transcription factor overexpression group versus the interference group, four vectors (pIRES2-TFDP1, pIRES2-TFDP1-NC, PLKO.1-PURO-TFDP1 and PLKO.1-PURO-TFDP1-NC) were constructed for cell transfection, and three biological replicates were used for each group. The *TFDP1* CDS of Yunshang black goats was amplified and cloned into the pIRES2-EGFP (Sangon Biotech, Shanghai, China) vector, resulting in the overexpression vector pIRES2-EGFP-TFDP1. TFDP1 was cloned into the PLKO.1-PURO (ampicillin) vector via *AgeI* and *EcoRI* to construct the *TFDP1* interfering plasmid (PLKO.1-PURO-TFDP1) in Yunshang black goats (GENEWIZ, Suzhou, China). The plasmid was extracted with a QIAGEN^®^ Plasmd Midi Kit (QIAGEN, Dusseldorf, Germany).

### 2.9. Granulosa Cell Isolation and Culture In Vitro

Primary ovarian granulosa cells (GCs) were isolated from the ovaries of goats according to the method described by Du et al. [43]. The goats were processed by estrus synchronization with a controlled internal drug releasing plug (CIDR, progesterone 300 mg, Inter Ag Co., Ltd., Auckland, New Zealand) for 12 d. Within 45–48 h (follicular phase, FP) after CIDR removal, the goats were sacrificed. After euthanization, ovarian samples were immediately collected and brought back to the laboratory to collect granulosa cells. Healthy goat ovaries were collected from the farm and stored in PBS containing a 1% penicillin–streptomycin solution (100 IU/mL penicillin and 50 mg/mL streptomycin) at 4 °C for transport to the laboratory. The ovaries were first washed three times with 75% alcohol, and excess tissue was cut away; the ovaries were then rinsed three times with PBS containing a 1% penicillin–streptomycin solution. Follicular fluid was extracted by puncturing >5 mm follicles with a 10 mL syringe placed in DMEM/F12 medium containing a 1% penicillin–streptomycin solution and 10% foetal bovine serum, and centrifuged at 1000 r/min for 8 min. The supernatant was discarded, and the cells were precipitated. Cells were resuspended in medium, inoculated on culture plates at a density of 2 × 10^5^ cells/mL and cultured at 37 °C under 5% CO_2_. Goat granulosa cells were extracted from >5 mm follicles, which represents the follicular phase. After the isolated granulosa cells had grown across the culture plates, they were passaged and used for subsequent experiments. The tests were conducted using GCs of up to five generations, which were still proliferating [44]. The FSHR and Vimentin (1:500; Bioss, Beijing, China) immunofluorescence assay was used to determine that the isolated cells were granulosa cells (shown in Appendix A Appendix A). The results of the immunofluorescence assay showed that over 95% of granulosa cells expressed FSHR and could be used in subsequent assays. The cells were incubated in 6-cm plates for 48 h (during which time the culture medium was changed once to ensure cell survival) and transferred to 10-cm plates for subsequent experiments. Goat ovarian granulosa cells and HEK293T cells (purchased from COBIOER, Shanghai, China) were cultured at 37 °C, 5% CO_2_, and 95% humidity. Ovarian granulosa cells were cultured in DMEM/F-12 medium (GIBCO, Thermo Fisher, Waltham, MA, USA), and HEK293T cells were cultured in DMEM (GIBCO, Thermo Fisher, Waltham, MA, USA); both were supplemented with 10% foetal bovine serum (GIBCO, Thermo Fisher, Waltham, MA, USA) and 100 U/mL penicillin/streptomycin (GIBCO, Thermo Fisher, Shanghai, China). The HEK293T cell line was thawed and transfected at 1 × 10^5^ cells/well in 24-well plates for dual luciferase reporter gene assays. Ovarian granulosa cells were transfected at 5 × 10^6^ cells/well in six-well plates to measure gene expression. Goat *TFDP1* and proliferation factor primers based on the NCBI database were used for RT–qPCR (Table 3).

### 2.10. Cell Transfection and Dual-Luciferase Reporter Assay

The plasmids were transfected into HEK293T cell lines using Lipofectamine 2000 (Invitrogen, Waltham, MA, USA) according to the manufacturer’s instructions. First, HEK293T cells were seeded on 24-well plates, and transfection was started after reaching 60–80% confluence. Diluted Lipofectamine 2000 reagent was then added to Opti-MEM medium and mixed. Master mixtures of different construct vectors were prepared by diluting the construct vectors in Opti-MEM medium and mixing well. The diluted construct vector was added to each tube of diluted Lipofectamine 2000 reagent (1:1) and incubated for 25 min. Finally, the construct vector-lipid complexes were added to the cells. Forty-eight hours after transfection, the cells were harvested and processed to measure luciferase activity. The remaining steps were performed according to the protocol of the Dual-Glo^®^ Luciferase Assay System (www.promega.com/protocols, TM058, USA, accessed on 15 February 2022).

The isolated cultured goat granulosa cells were inoculated in 96-well plates at 5 × 10^4^ cells/well and cultured in DMEM/F12 medium containing 1% penicillin–streptomycin and 10% FBS for 24 h. The Opti-MEM medium was replaced, and the plasmids were transfected into the granulosa cells using Lipofectamine 2000. The liquid was removed and replaced with the original medium again, and subsequent experiments were performed. Proliferation experiments were performed with three replicates per group. A vector with GFP labelling was used to detect the efficiency of transfection, and the results showed that the cell transfection efficiency reached 70–80% (shown in Appendix A Appendix A).

### 2.11. Cell Counting Kit-8 (CCK-8) and EdU Assays

Goat primary GCs were cultured in 96-well plates. Cell proliferation was detected after transfection with a CCK-8 Cell Proliferation Assay Kit (Sollerbauer Technology, Beijing, China). After transfection treatment of the cells, the culture medium was changed after 6 h, and 10 μL of CCK-8 reagent was added to each well. The cells were incubated for 2 h, and the optical density values were detected at 450 nm with a microplate reader (Thermo Fisher, Varioskan LUX, MA, USA) at 0, 6, 12, 24 and 48 h.

The CBeyoClick™ EdU-488 Cell Proliferation Assay Kit (Beyotime Biotechnology, Beijing, China) was used to determine the proliferation status of goat ovarian granulosa cells according to the manufacturer’s guidelines. Briefly, 6-well plates were used to add EdU at a final concentration of 10 μM per well after transfecting pIRES2-TFDP1 with PLKO.1-PURO-TFDP1 during 6 h of incubation. The cells were washed three times with PBS (Thermo Fisher, Waltham, MA, USA), fixative was added for 15 min at room temperature and then removed, and the cells were rinsed three times with a washing solution. The cells were incubated with 1 mL of 0.3% Triton X-100 PBS for 15 min at room temperature to increase the permeability of the cell membrane. In addition, 500 μL of the prepared Click reaction solution was added to each well. The cells were incubated for 30 min at room temperature (protected from light) to observe and quantify the number of cells stained with EdU. Three regions of view were randomly selected for statistical analysis.

## 3. Results

### 3.1. DNA and RNA Quality Measurements and SNP Analysis

DNA (Figure 1A) and total RNA (Figure 1B) were extracted and measured for mass and concentration, and PCR products were identified by electrophoresis (Figure 1C), all of which were consistent with subsequent experiments. The SNP locus was identified using DNAstar, and according to the annotation of the NCBI database GenBank NC_030833.1, the locus described is located at locus g.36491960 on goat chromosome 26. The SNP is located in the RBP4 5′ flanking region −211 bp, the locus is not publicly available and has taken the HGVS nomenclature as *RBP4* g.36491960G>C (Figure 1D).

### 3.2. RBP4 Polymorphism Analysis

The results of *RBP4* g.36491960G>C genotyping of the 400-goat population showed 150 GG, 174 GC and 54 CC genotypes (Figure 2). The population genetics analysis showed that the g.36491960G>C locus had a polymorphic information content of 0.358, which was moderately polymorphic (0.25 < *PIC* < 0.50) (Table 4). The chi-square test results indicated that the g.36491960G>C group was in Hardy-Weinberg equilibrium (chi-square test criterion, 0.757; *p* > 0.05).

### 3.3. Association Analysis of SNPs with Goat Reproductive Traits

To further investigate the effect of *RBP4* g.36491960G>C on reproductive traits in Yunshang black goats, an independent samples t-test was used to analyse the association between g.36491960G>C and the first, second and third kidding and the mean number of kids. *RBP4* g.36491960G>C was highly significantly associated with the number of first kids in Yunshang black goats (*p* < 0.01), and the number of kids in individuals with the CC genotype was significantly higher than those with the GC and GG genotypes (*p* < 0.05) (Table 5). Both the CC and GC genotypes had a significantly higher number of kids born in the second litter than the GG genotype (*p* < 0.05). The number of kids born in the third litter and the mean number of kids born to individuals with the CC and GC genotypes were both highly significantly higher those in the GG genotype (*p* < 0.01); there were no significant differences between the GC and CC genotypes. These results suggested that the g.36491960G>C polymorphism in the *RBP4* promoter region was involved in the regulation of the kidding number in Yunshang black goats.

### 3.4. The g.36491960G>C Polymorphism Affected the Promoter Activity of RBP4

To determine the effect of g.36491960G>C on the *RBP4* promoter region, pGL3-basic, pGL3-C and pGL3-G were transfected into HEK293T cells. The dual luciferase reporter assay showed that the luciferase activity of plasmid pGL3-C was significantly higher than those of pGL3-G and pGL3-basic (*p* < 0.05) (Figure 3A). The *RBP4* haplotype RT–qPCR results showed that expression of the *RBP4* CC genotype was significantly higher than that of the GG genotype (Figure 3B). The EMSA results showed that individuals with the *RBP4* g.36491960G>C CC genotype had a significantly higher promoter binding capacity than that with the GG genotype (Figure 3C). These results suggested that individuals with the C allele of the *RBP4* promoter region have higher transcriptional activity than individuals with the G allele. In addition, binding sites for many transcription factors such as E2F6, TFDP1, E2F1 and E2F4 were predicted to be present in this region.

### 3.5. The g.36491960G>C Mutation Adds a Novel Transcription Factor-Binding Site

Prediction of the *RBP4* g.36491960G>C polymorphic site combined with transcription factors using the online tool JASPAR revealed the highest score for the newly added transcription factor TFDP1 at this locus (Figure 4A). To further investigate the effect of TFDP1 on the kidding number of Yunshang black goats, the RT–qPCR results of ovarian tissues without kidding number showed that the expression of TFDP1 in the ovarian tissues of goats in the high-prolificacy group was significantly higher than that in the low-prolificacy group (*p* < 0.05) (Figure 4B).

### 3.6. TFDP1 Is Involved in Regulating the Expression of RBP4

To verify the role of TFDP1 in regulating *RBP4* expression, the plasmids pIRES2-TFDP1 and PLKO.1-PURO-TFDP1 were transfected into goat granulosa cells. The RT–qPCR results showed that TFDP1 expression levels were significantly increased or decreased after overexpression and interference with TFDP1, respectively, in goat granulosa cells (*p* < 0.05) (Figure 5A,B). The expression of *RBP4* was significantly increased after overexpression of TFDP1 (*p* < 0.05) and significantly decreased after interference with TFDP1 (*p* < 0.01) (Figure 5C,D).

### 3.7. Regulation of GC Proliferation by the Transcription Factor TFDP1

To further verify the effect of the transcription factor TFDP1 on the proliferation of goat GCs, the expressions of the cell proliferation factors cyclin D1, cyclin D2 and CDK4 were quantified. The RT–qPCR results showed that the expressions of the cell proliferation factors cyclin D1, cyclin D2 and CDK4 were positively correlated with the expression of TFDP1. Cell proliferation factor expression was significantly upregulated after overexpression of TFDP1 in goat granulosa cells, whereas interference with TFDP1 resulted in significant downregulation of cell proliferation factors (*p* < 0.05 or *p* < 0.01) (Figure 6A,B). Both the CCK-8 and EdU results showed that GCs overexpressing TFDP1 had a higher proliferation rate than that of the control group (Figure 6C,E). Similarly, the proliferation rate of TFDP1-silenced GCs was lower than that of the control group (Figure 6D,E). All these results suggested that the transcription factor TFDP1 has a promotional effect on the proliferation of goat GCs.

## 4. Discussion

Granulosa cells (GCs) are known to proliferate in the follicles of female ovarian tissue and are important somatic cells surrounding the oocyte [45]. The main roles of ovarian granulosa cells are to transport nutrients to the oocytes, convert androgens into estrogens and synthesize progesterone. Ovarian granulosa cells undergo several biochemical processes during folliculogenesis and are involved in the development of ovarian follicles, which has a major impact on reproduction [46]. Retinol-binding protein 4 (RBP4), a member of the RBP protein family, is expressed in the uterus and embryo in the early stages of pregnancy [47]. *RBP4* directs embryonic development by binding to retinol [48]. Available studies have shown that RBP4 is highly associated with ovarian diseases such as PCOS [27], human ovarian cancer [49,50,51,52] and porcine ovarian cysts [53]. It has also been shown that polymorphisms in *RBP4* can significantly affect backfat thickness [54] and foetal size in pigs [55]. In this study, to investigate the effect of *RBP4* on goat fertility, a mutation in the promoter region of *RBP4* g.36491960G>C in Yunshang black goats was identified by PCR amplification. An association analysis with kidding number revealed that the mutation significantly increased the number of Yunshang black goats born in the first, second and third litters. These results suggest that *RBP4* might be a candidate molecular marker for breeding high fertility breeds such as Yunshang black goats. Furthermore, a comparison of expression in the ovaries of *RBP4* homozygous individuals with different genotypes showed that expression was significantly higher in individuals with the CC genotype than those with the GG and GC genotypes. This suggests that *RBP4* is an important candidate gene associated with high fertility traits in goats. Previously, no association between reproduction and *RBP4* has been reported in studies of reproductive traits in goats or sheep. However, in studies of candidate genes associated with high fertility in pigs, *RBP4* SNPs were found to significantly increase total litter size [56]. For example, Rothschild et al. studied six commercial lines of pigs and found that the *RBP4* polymorphism had a significant effect on the total number of offspring per litter. There are many other studies showing a relationship between *RBP4* polymorphisms and litter size [57,58,59,60,61]. In addition, it has also been found that *RBP4* can enhance the proliferation and invasion of HTR8 cells by inhibiting PI3K/AKT signalling [62]. This result suggests that *RBP4* may also play a role in promoting cell growth and development to some extent.

Gene expression is a complex process influenced by several biological regulatory elements, among which transcription initiation is essential for gene expression [63]. Transcription factors play an important role as eukaryotic transcriptional initiation elements. To further investigate the effect of mutations in the 5′ flanking region of *RBP4* on goat reproductive traits, transcription factor prediction revealed that the *RBP4* g.36491960G>C mutation added a new binding site for the transcription factor TFDP1. *TFDP1* is a member of the TFDP family [64]. Under normal conditions, the TFDP family plays a key role in a variety of essential life processes by interacting with the E2F family [65]. The E2F family plays an important role in early cell cycle growth by controlling the up- or downregulation of many other genes in the cell cycle pathway to stimulate or inhibit cell growth [66,67,68,69]. Although TFDP1 mainly binds to E2F1 to form a TFDP1/E2F1 heterodimer, it was found that this heterodimer is associated with the cyclic regulation of cellular genes. These genes are essential for passage through the G1 phase and into the S phase [68,70]. It has also been shown that *TFDP1* deletion leads to extraembryonic developmental defects and lethality [71]. Thus, we speculate that the altered promoter activity caused by the RBP4 g.36491960G>C mutation may be related to the transcription factor-binding site for the newly generated *TFDP1*. Therefore, we constructed overexpression and interference vectors for the transcription factor TFDP1 and then transfected them into goat GCs separately to assay *RBP4* expression and GC proliferation. The results showed that the expression of *TFDP1* was positively correlated with the expression of *RBP4*. Our results indicate that the transcription factor *TFDP1* can influence the proliferation of goat GCs by affecting the expression of *RBP4*. This is consistent with the finding of Elis et al. that *RBP4* is able to induce the proliferation of bovine GCs [72]. Although goats and cattle are different species, their genomes are highly homologous, so they may provide scientific support for the development of this study. In addition, in other mammals, *RBP4* has been shown to promote mitosis in GCs [73,74]. These studies further corroborate our results. Briefly, the transcription factor *TFDP1* was able to influence the proliferation of goat GCs through the regulation of *RBP4*.

## 5. Conclusions

In summary, the process of gene transcriptional expression is very complex and remains largely unknown. Our study provided a new perspective: a g.36491960G>C mutation in the 5′ flanking region of *RBP4* was significantly associated with the kidding number of Yunshang black goats and resulted in increased *RBP4* promoter activity. This mutation creates a new binding site for the transcription factor *TFDP1*, which increases *RBP4* promoter activity and influences the proliferation of GCs in goats (Figure 7). This study provides a new perspective for goat breeding.

## Figures and Tables

**Figure 1 cells-11-02148-f001:**
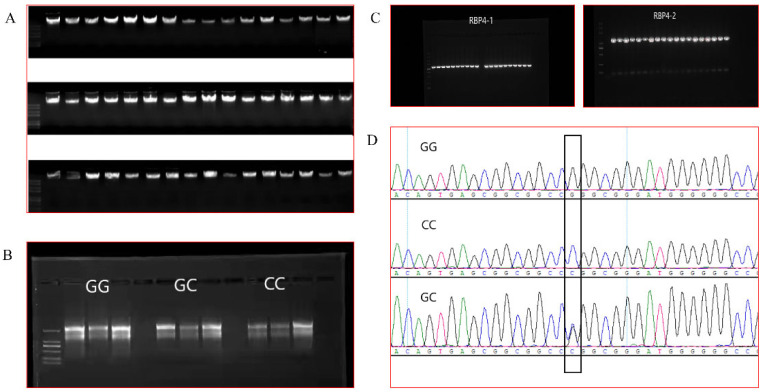
(**A**): Partial DNA electrophoresis results. (**B**): Electrophoresis results of total RNA from individuals of the three genotypes. (**C**): Electrophoresis results of PCR products. (**D**): Sequencing results for the *RBP4* g.36491960G>C locus.

**Figure 2 cells-11-02148-f002:**
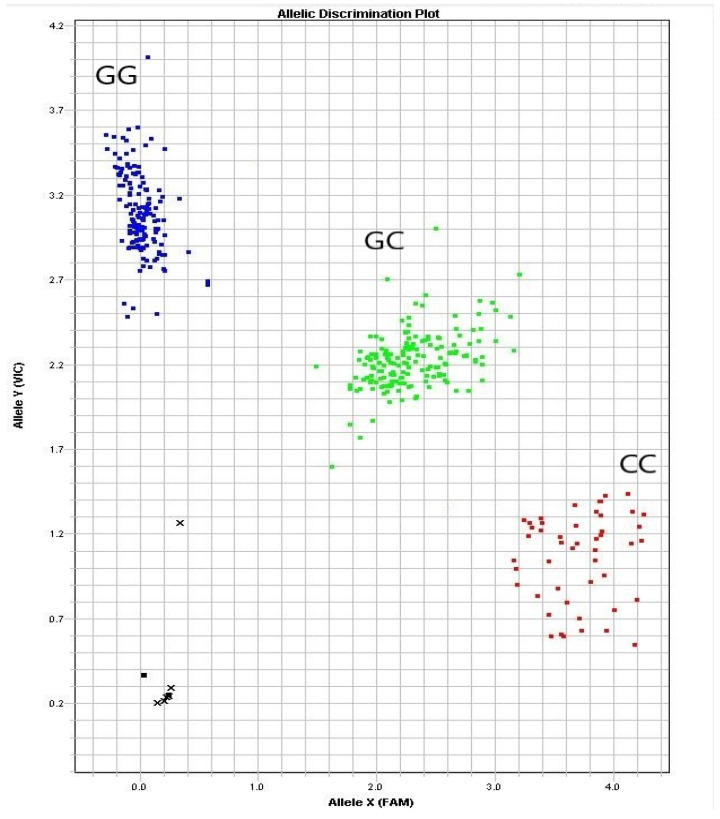
Discriminant map of the three clusters of alleles of the *RBP4* g.36491960G>C mutation.

**Figure 3 cells-11-02148-f003:**
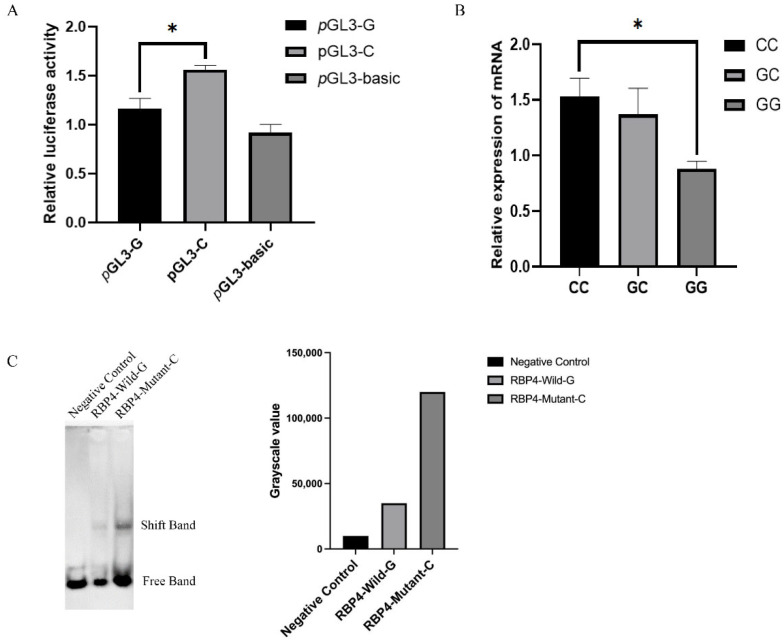
The g.36491960G>C mutation regulated the promoter activity of *RBP4*. (**A**): The result of the dual luciferase activity assay. (**B**): Expression levels of RBP4 in different genotypes. (**C**): EMSA of RBP4 individuals with different genotypes. The results were expressed as the mean ± SEM (*n* = 3). * *p* < 0.05.

**Figure 4 cells-11-02148-f004:**
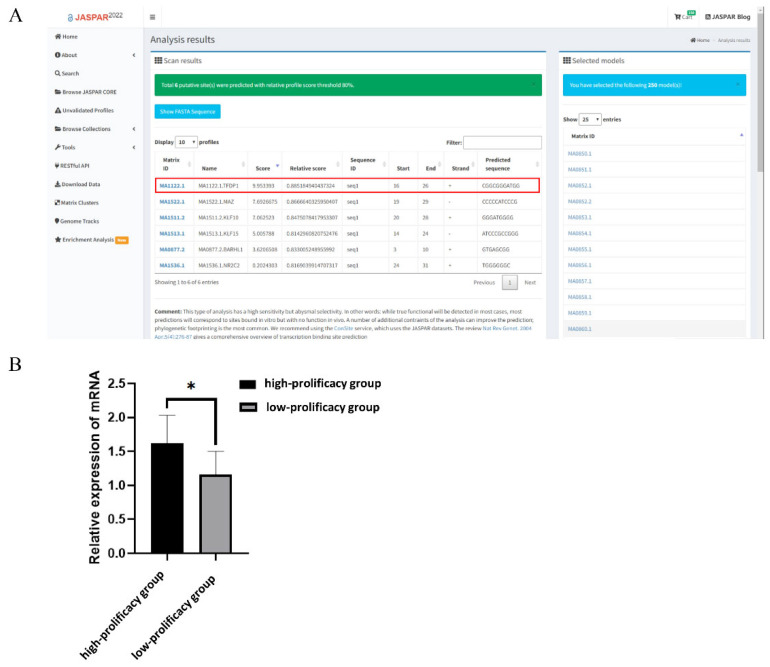
(**A**): *RBP4* g.36491960G>C mutation binding transcription factor prediction (URLs https://jaspar.genereg.net/, accessed on 5 January 2022). (**B**): Expression of TFDP1 in ovarian tissues of Yunshang black goats in the high-prolificacy and low-prolificacy groups. The results were expressed as the mean ± SEM (*n* = 3). * *p* < 0.05.

**Figure 5 cells-11-02148-f005:**
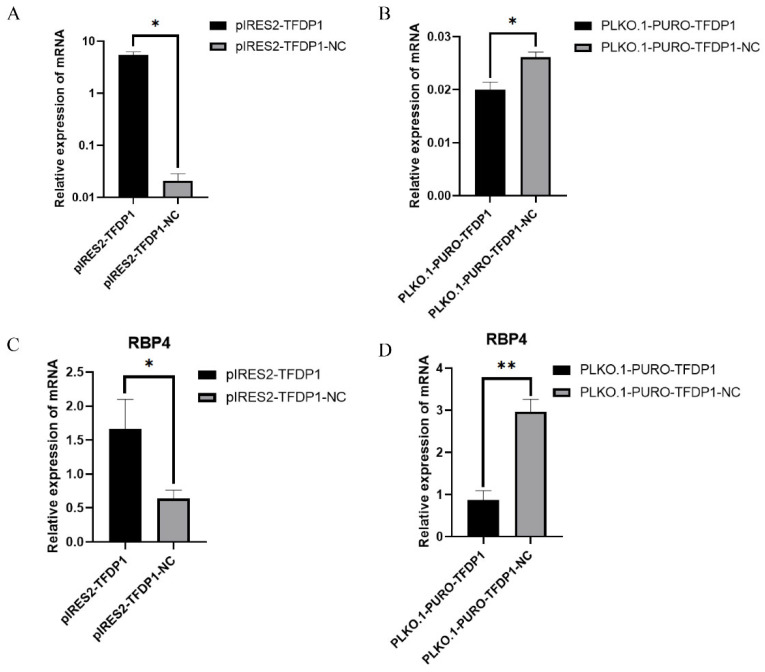
(**A**,**B**): The expression of *TFDP1* in goat GCs after TFDP1 overexpression and interference. (**C**,**D**): The expression of *RBP4* in goat GCs after TFDP1 overexpression and interference. The results were expressed as the mean ± SEM (*n* = 3). * *p* < 0.05; ** *p* < 0.01.

**Figure 6 cells-11-02148-f006:**
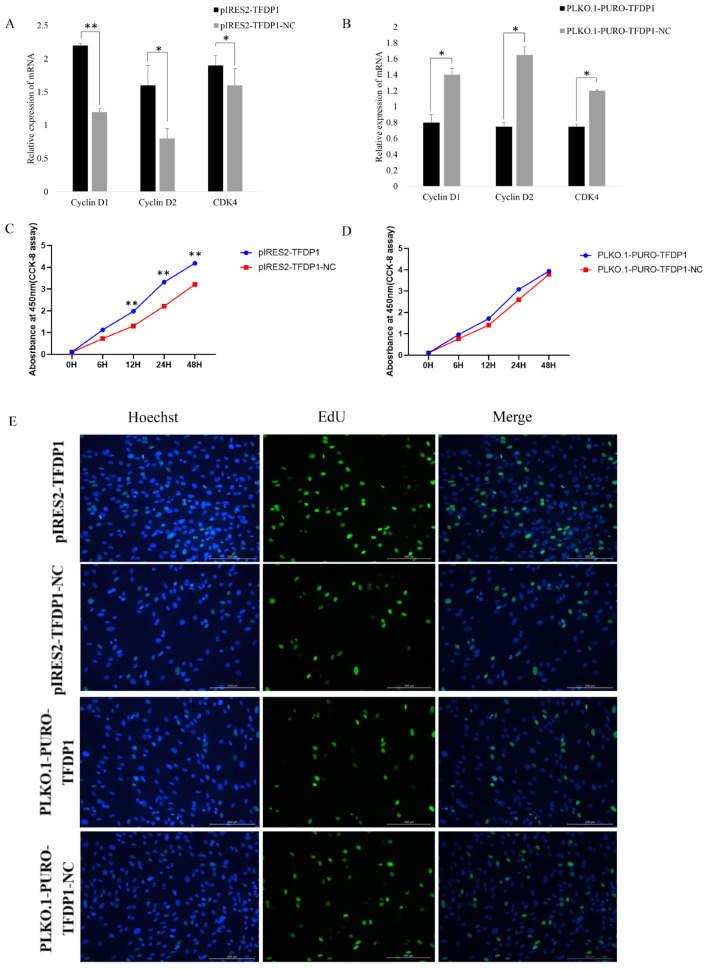
The transcription factor *TFDP1* regulated the proliferation of goat GCs. (**A**,**B**): The expression of proliferative factors after *TFDP1* overexpression and interference, respectively, in goat GCs. (**C**,**D**): The CCK-8 results. (**E**): The EdU assay of GC proliferation. * *p* < 0.05; ** *p* < 0.01.

**Figure 7 cells-11-02148-f007:**
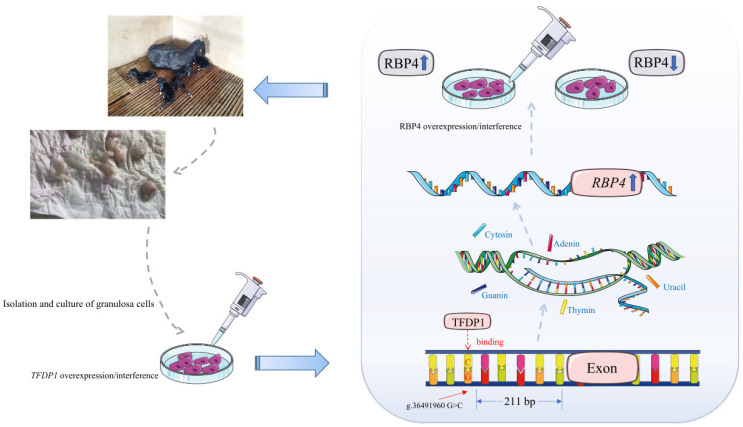
Flow chart of the *RBP4* g.36491960G>C mutation regulating the proliferation of goat granulosa cells by the transcription factor TFDP1.

**Table 1 cells-11-02148-t001:** Primers for the *RBP4* promoter region.

Gene	Accession Number	Primer Sequence (5′–3′)	Tm (°C)	Product Length (bp)
*RBP4*-F1	NC_030833.1	AGGTGGAAAGTGAAAGCGG	60	1183
*RBP4*-R1		AGAACGAGGGACATCTGCG		
*RBP4*-F2		ATCGCAGATGTCCCTCGT	60	1226
*RBP4*-R2		CGTCCCAGTTACTGCGAA		

**Table 2 cells-11-02148-t002:** Primers and probes for SNP-KASP.

Name	Accession Number	Primer Sequence (5′–3′)
Primer_AlleleFAM	NC_030833.1	GAAGGTGACCAAGTTCATGCTCTCCCGACAGTGAGCGGCGGCCG
Primer_AlleleHEX		GAAGGTCGGAGTCAACGGATTCTCCCGACAGTGAGCGGCGGCCC
Primer_Common		CGGTCCCCAGGCTCCATCTTGCC

**Table 3 cells-11-02148-t003:** Information on the primers used for RT–qPCR.

Gene Name	Primer Sequence (5′–3′)	Tm (°C)	Product Length (bp)	GenBank Number
*RBP4*-F	TAAATAACTGGGACGTGTGTGC	60	127	NC_030833.1
*RBP4*-R	ATCCAGTGGTCATCGTTTCCT			
*RPL19*-F	ATCGCCAATGCCAACTC	60	154	NC_030826.1
*RPL19*-R	CCTTTCGCTTACCTATACC			
*TFDP1*-F	TGACAGAAATGGCTCAGGGTT	60	223	NW_017189962.1
*TFDP1*-R	TCCTCGTTCTCGTTGAAGTCC			
*Cyclin D1*-F	GCCACAGACGTGAAGTTCATTT	60	156	NC_030836.1
*Cyclin D1*-R	CGGGTCACATCTGATCACCTT			
*Cyclin D2*-F	ATGTGGATTGCCTCAAAGCC	60	152	NC_030812.1
*Cyclin D2*-R	CAGGTCGATATCCCGAACATC			
*CDK4*-F	GAGCATCCCAATGTTGTCAGG	60	172	NC_030812.1
*CDK4*-R	ACTGGCGCATCAGATCCTTT			

**Table 4 cells-11-02148-t004:** Population polymorphism analysis of loci in goats.

Locus	Genotype Frequency			Allele Frequency		*PIC*	*He*	*Ne*	*p* Value
	GG	GC	CC	C	G				
*RRB4* g.36491960G>C	0.397	0.460	0.143	0.373	0.627	0.358	0.468	1.879	0.757

**Table 5 cells-11-02148-t005:** Least-squares means and standard errors of kidding number in Yunshang black goats with different genotypes.

Locus	Genotype	Number	1st Parity Kidding Number	2nd Parity Kidding Number	3rd Parity Kidding Number	Average Kidding Number
*RBP4* g.36491960G>C	GG	150	1.893 ± 0.051 ^a^	2.027 ± 0.064 ^a^	2.107 ± 0.061 ^a^	1.966 ± 0.042 ^a^
	GC	174	2.052 ± 0.047 ^b^	2.206 ± 0.057 ^b^	2.341 ± 0.056 ^b^	2.160 ± 0.039 ^b^
	CC	54	2.167 ± 0.085 ^b^	2.341 ± 0.104 ^b^	2.529 ± 0.107 ^b^	2.189 ± 0.070 ^b^

Note: Different lowercase letters in the same group indicate significant differences (*p* < 0.05).

## Data Availability

Not applicable.

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
