# Peer review of "Effect of Upregulation of Transcription Factor TFDP1 Binding Promoter Activity Due to RBP4 g.36491960G>C Mutation on the Proliferation of Goat Granulosa Cells"

_cells, 2022, doi:10.3390/cells11142148_

Round 1
Reviewer 1 Report
The authors have performed comments, thus the manuscript can be accepted for publication.
Author Response
Thank you very much for the time and effort you put into our manuscripts.
Reviewer 2 Report
1. Figure 4 (A), for prediction of transcription factors, it is recommended to list the predicted URLs used by the authors.
2. For acronyms that appear for the first time, the full name needs to be listed, and subsequent writings only need to write abbreviations. (e.g.L392). The author also needs to revise the whole article carefully.
3. L408-410, at least one reference is missing here.
4. L446 “is” need revise to “was”
5. Spelling mistake: Supplementary Materials S1 “Marge”?
6. The author needs to revise the language of the full text. For example, word spelling, grammar, tenses, etc.
Author Response
Thank you very much for evaluating our work! We have tried our best to improve the manuscript according to the reviewer’s further comments.
- Figure 4 (A), for prediction of transcription factors, it is recommended to list the predicted URLs used by the authors.
Response: Thank you. We have added the URLs in the new version (line 347-348).
- For acronyms that appear for the first time, the full name needs to be listed, and subsequent writings only need to write abbreviations. (e.g.L392). The author also needs to revise the whole article carefully.
Response: Thank you. We have revised in the new version.
- L408-410, at least one reference is missing here.
Response: Thank you. We have added in the new version (line 404).
- L446 “is” need revise to “was”
Response: Thank you. We have revised in the new version (line 437).
- Spelling mistake: Supplementary Materials S1 “Marge”?
Response: Thank you. We have revised in the new version.
- The author needs to revise the language of the full text. For example, word spelling, grammar, tenses, etc.
Response: Thank you very much! The language has been polished by the AJE company (https://www.aje.cn/).
This manuscript is a resubmission of an earlier submission. The following is a list of the peer review reports and author responses from that submission.
Round 1
Reviewer 1 Report
Please see comments given in the reviewed attached file of manuscript.

Author Response
Thank you very much for evaluating our work! We have tried our best to improve the manuscript according to the reviewer’s further comments.
- Please add reference for this sentence. for this you can use below reference: Askari N, Mohammadabadi M, Baghizadeh A 2011. ISSR markers for assessing DNA polymorphism and genetic characterization of cattle, goat and sheep populations. Iranian Journal of Biotechnology 9 (3), 222-229.
Response: Thank you for your advice. It was added in the new version (line 44).
- You refer to references that mostly belong to china. it is better to refer to studies from other countries.
for this you can use below references:
Alinaghizadeh H, Mohammad Abadi MR, Zakizadeh S 2010. Exon 2 of BMP15 gene polymorphismin Jabal Barez Red Goat. Journal of Agricultural Biotechnology 2 (1), 69-80.
Mohammadabadi MR 2021. Tissue-specific mRNA expression profile of ESR2 gene in goat. Agricultural Biotechnology Journal 12 (4), 169-184.
Gooki FG, Mohammadabadi M, Fozi MA, Soflaei M 2019. Association of Biometric Traits with Growth Hormone Gene Diversity in Raini Cashmere Goats. Walailak Journal of Science and Technology (WJST) 16 (7), 499-508.
Gholamhoseini G F, Mohammadabadi MR, Asadi Fozi M 2018. Polymorphism of the growth hormone gene and its effect on production and reproduction traits in goat. Iranian Journal of Applied Animal Science 8 (4), 653-659.
Response: Thank you very much. We have replaced the references in the new version (line 52).
- It is better to explain about importance of genome studying. For this you can use below sentences and references:
Moreover, the epigenome comprising different mechanisms e.g. DNA methylation, remodeling, histone tail modifications, chromatin microRNAs and long non-coding RNAs, interact with environ-mental factors like nutrition, pathogens, climate to influence the expression profile of genes and the emergence of specific pheno-types (Barazandeh et al. 2019; Masoudzadeh et al., 2020a). Multi-level interactions between the genome, epigenome and environmental factors might occur (Mohamadipoor et al., 2021). Furthermore, numerous lines of evidence suggest the influence of epigenome variation on health and production (Barazandeh et al. 2019; Mohammadabadi et al. 2017; Masoudzadeh et al., 2020b). The expression of eukaryotic genes is temporarily and multidimensionally controlled (Shahsavari et al., 2021). Only a relatively small set of the entire genome is expressed in each type of tissue, and the expression of genes depends on the stage of development (Tohidi Nezhad et al., 2015). Therefore, gene expression in eukaryotes is specific to each tissue (Mohammadabadi et al., 2021). Also, the amount of gene products that are made in the same tissue as well as in other tissues that make up that product, regulates the expression of that gene (Mohammadabadi, 2019). One of the basic activities in domestic animals is the study of genes and proteins related to economic traits and their study at the cellular or chromosomal level (Mohammadabadi and Asadollahpour Nanaei, 2021).
Barazandeh A, Mohammadabadi MR, Ghaderi-Zefrehei M, Rafeied F, Imumorin IG 2019. Whole genome comparative analysis of CpG islands in camelid and other mammalian genomes. Mammalian Biology 98, 73-79
Masoudzadeh, S.H., Mohammadabadi, M., Khezri, A., Stavetska, R.V., Oleshko, V.P., Babenko, O.I., Yemets, Z., Kalashnik, O.M., 2020b. Effects of diets with different levels of fennel (Foeniculum vulgare) seed powder on DLK1 gene expression in brain, adipose tissue, femur muscle and rumen of Kermani lambs. Small Ruminant Research 193, e106276.
Masoudzadeh, S.H., Mohammadabadi, M.R., Khezri, A., Kochuk-Yashchenko, O.A., Kucher, D.M., Babenko, O.I., Bushtruk, M.V., Tkachenko, S.V., Stavetska, R.V., Klopenko, N.I., Oleshko, V.P., Tkachenko, M.V., Titarenko, I.V., 2020a. Dlk1 gene expression in different Tissues of lamb. Iranian Journal of Applied Animal Science 10, 669-677.
Mohammadabadi M.R., Jafari A.H.D. and Bordbar F. (2017). Molecular analysis of CIB4 gene and protein in Kermani sheep. Brazil. J. Medic. Biol. Res.50, e6177.
Shahsavari M, Mohammadabadi M, Khezri A, Asadi Fozi M, Babenko O, Kalashnyk O, Oleshko V, Tkachenko S 2021. Correlation between insulin-like growth factor 1 gene expression and fennel (Foeniculum vulgare) seed powder consumption in muscle of sheep. Animal Biotechnology, 1-11.
Mohamadipoor L, Mohammadabadi M, Amiri Z, Babenko O, Stavetska R, Kalashnik O, Kucher D, Kochuk-Yashchenko O, Asadollahpour H 2021. Signature selection analysis reveals candidate genes associated with production traits in Iranian sheep breeds. BMC veterinary research 17 (1), 1-9
Mohammadabadi M, Masoudzadeh SH, Khezri A, Kalashnyk O, Stavetska RV, Klopenko NI, Oleshko VP, Tkachenko SV 2021. Fennel (Foeniculum vulgare) seed powder increases Delta-Like Non-Canonical Notch Ligand 1 gene expression in testis, liver, and humeral muscle tissues of growing lambs. Heliyon 7 (12), e08542.
Mohammadabadi, M.R., 2019. Expression of calpastatin gene in Raini Cashmere goat using Real-Time PCR. Agricultural Biotechnology Journal 11, 219-235.
Mohammadabadi, M.R., Asadollahpour Nanaei, H., 2021. Leptin gene expression in Raini Cashmere goat using Real-Time PCR. Agricultural Biotechnology Journal 13, 197-214.
Response: Thank you very much. We have added the sentences in the new version (line 79-92).
- How? After slaughtering or using biopsy? please identify in the text of manuscript
Response: Thank you for your valuable and important advice. Ovarian tissue were obtained from the slaughtered female goats, and all the experiment animals were euthanized. We have revised in the new version (line 110-112).
- Please add accession number for used primers
Response: Thank you. We have added the accession number in the new version (Table 1).
- Please add accession number for used primers
Response: Thank you. We have added the accession number in the new version (Table 2).
- Please add reference
Response: Thank you. We have added the reference in the new version (line 173).
- conclusions are repeat of your results. it is better to improve it based on your results.
Response: Thank you. We have rephased the conclusion part in the new version (line 439-445).
Reviewer 2 Report
This is a study in goat, addressing the role of Retinol-binding protein 4 (RBP4) and TFDP1 in ovarian granulosa cell function and proliferation. The study appears, in most parts, at least at first glance, to be well conducted and the text reads also well.
There are, however, upon a closer look a few , yet very important points that remain unclear to me and missing information:
1. The authors refer to a previous study Du et al., (ref# 32) when it comes to the description of goat granulosa cell isolation and culture. They should in addition, at least briefly describe this method. From which follicular stage were the cells taken? When were they used for the experiments? Maybe goat is different in this aspect, but usually when granulosa cells are isolated from follicles they rapidly luteinize and then they do not proliferate any more. Are the author sure that they cultured and then studied granulosa cells? How was this controlled and pure were the cultures? How was this determined? This is crucial - missing- information!
2. Also, I miss information about the exact transfection procedures of goat granulosa cells. I am not aware that - in general- granulosa cells can be readily transfected and therefore I am a bit surprised about this data in goat. The whole procedure, the efficiency and also (missing!) control experiments (no plasmid) need to be described in much more detail. Also, how often were the proliferation assays reated (not indicated in Fig. 6)?
Without such information the authors can not rightly claim a role in proliferation of these ovarian cells, I am afraid.
Author Response
Thank you very much for evaluating our work! We have tried our best to improve the manuscript according to the reviewer’s further comments.
- The authors refer to a previous study Du et al., (ref# 32) when it comes to the description of goat granulosa cell isolation and culture. They should in addition, at least briefly describe this method. From which follicular stage were the cells taken? When were they used for the experiments? Maybe goat is different in this aspect, but usually when granulosa cells are isolated from follicles they rapidly luteinize and then they do not proliferate any more. Are the author sure that they cultured and then studied granulosa cells? How was this controlled and pure were the cultures? How was this determined? This is crucial - missing- information!
Response: Thank you for your valuable and important comments. Healthy goat ovaries were collected from the farm and stored in PBS containing 1% penicillin-streptomycin solution (100 IU/mL penicillin and 50 mg/mL streptomycin) at 4℃ for transport to the laboratory. Ovaries were firstly washed 3 times with 75% alcohol, excess tissue was cut away, and then rinsed 3 times with PBS containing 1% penicillin-streptomycin solution. Follicular fluid was extracted by puncturing >5 mm follicles with a 10 mL syringe and placed in DMEM/F12 medium containing 1% penicillin-streptomycin solution and 10% fetal bovine serum, centrifuged at 1000 r/min for 8 min, the supernatant was discarded and cells were precipitated. Cells were resuspended with medium and then inoculated on culture plates at a density of 2×105 cells/mL and cultured at 37℃ with 5% CO2. Goat granulosa cells were extracted from >5 mm follicles, which was at the follicular phase. After the isolated granulosa cells had grown all over the culture plates, they were passaged and could be used for subsequent experiments. The test was carried out using granulosa cells of up to 5 generations, which were still proliferating. The FSHR immunofluorescence assay is used to verify that the isolated cells were granulosa cells (shown in supplementary Figure S1). The result of immunofluorescence assay showed that over 95% of granulosa cells expressed the FSHR and could be used in subsequent assays. We hope that these contents had answered your questions well and had added the information in the new version (lines 213-229).
- Also, I miss information about the exact transfection procedures of goat granulosa cells. I am not aware that - in general- granulosa cells can be readily transfected and therefore I am a bit surprised about this data in goat. The whole procedure, the efficiency and also (missing!) control experiments (no plasmid) need to be described in much more detail. Also, how often were the proliferation assays reated (not indicated in Fig. 6)?
Without such information the authors can not rightly claim a role in proliferation of these ovarian cells, I am afraid.
Response: Thank you for your valuable and important comments. The isolated cultured granulosa cells of goats were inoculated in 96-well plates at 5 × 104 cells/well and cultured in DMEM/F12 medium containing 1% penicillin-streptomycin solution and 10% FBS for 24 h. And the Opti-MEM medium was replaced and the vector was transfected into the granulocytes using Lipofectamine 2000. The liquid was removed, replaced again with the original medium and subsequent experiments were performed. Proliferation experiments were performed with 3 replicates per group. The vector with GFP label was used to detect the efficiency of transfection, and result showed that the cell transfection efficiency reached to 70%-80% (shown in supplementary material Figure S2). I am very sorry that a blank control without transfection of the plasmid was not designed in this study. In the recent time, the COVID-19 is very serious, and the goat granulosa cells were more difficult to collect. Therefore, a blank control was not added to the experiments. Fortunately, the results of the study so far were also as expected and next time we will be more rigorous in the experiments design. Thank you very much for the time and effort you had put into improving this study. We hope that these contents had answered your questions well and had added the information in the new version (lines 253-261).
Round 2
Reviewer 2 Report
When looking at the reivsed version, I am afraid that a main point raised earlier is not adequatly addressed - the assumed nature of granulosa cells. The authors now show a immunfluorescen image of FSHR, but there is no information about the antibody used and no control is provided.! Therefore I am not concinced that we are looking at granulosa cells in the first place. The authors must convincingly show their nature (e.g. aromatase expression; steroid production...). It is unclear why the cells studied proliferate in culture (to my knowledge isolated granulosa cells luteinize!) and this raises the question, whether the authors are rather looking at fibroblasts. Therefore, fibroblast markers must also be tested.